# Perfect circular polarization of elastic waves in solid media

Jeseung Lee[1], Minwoo "Joshua" Kweun [2] ✉, Woorim Lee[3],
Hong Min Seung [4,5] & Yoon Young Kim [1,3] ✉

Elastic waves involving mechanical particle motions of solid media can couple volumetric and shear deformations, making their manipulation more difficult than electromagnetic waves. Thereby, circularly polarized waves in the elastic regime have been little explored, unlike their counterparts in the electromagnetic regime, where their practical usage has been evidenced in various applications. Here, we explore generating perfect circular polarization of elastic waves in an isotropic solid medium. We devise a novel strategy for converting a linearly polarized wave into a circularly polarized wave by employing an anisotropic medium, which induces a so-far-unexplored coupled resonance phenomenon; it describes the simultaneous occurrence of the Fabry-Pérot resonance in one diagonal plane and the quarter-wave resonance in another diagonal plane orthogonal to the former with an exact 90° out-of-phase relation. We establish a theory explaining the involved physics and validate it numerically and experimentally. As a potential application of elastic circular polarization, we present simulation results demonstrating that a circularly polarized elastic wave can detect an arbitrarily oriented crack undetectable by a linearly polarized elastic wave.

As an elastic analogy of an optical birefringent plate, in 1964, Einspruch proposed a conceptual scheme for generating a circularly polarized elastic wave using tungsten buffers[1]. However, the classical birefringent plate has an intrinsic trade-off between its bulkiness and the accuracy of the circularly polarized elastic waves it generates. The elastic birefringent plate is typically bulky because elastic waves in solids have much longer wavelengths than electromagnetic waves. Strong birefringence is required to avoid bulkiness, but it inevitably sacrifices the accuracy of the generated circular polarization. The detailed theoretical analysis of the intrinsic limitations of the classical birefringent plate is described in Supplementary Note 1. Since the first identification of shear-wave splitting due to the seismic birefringence of the Earth's crust[2] and upper mantle[3] in 1980, the observation of elliptically or circularly polarized elastic waves in the anisotropic Earth has been studied for decades[4–6]. In addition, some researchers have

reported the realization of circularly polarized elastic waves in metals at low temperatures[7] and artificially structured chiral materials[8–12]. Despite these efforts, the utilization of circularly polarized elastic waves is still difficult due to the lack of a method to effectively generate elastic circular polarization in common isotropic solids, which constitute most load-carrying elements in engineering applications.

Here, we propose a novel resonance mechanism for the generation of perfect circular polarization of elastic waves in isotropic solids. Our approach is to perfectly convert an incoming linearly polarized elastic wave through an anisotropic medium (realized by a metamaterial) into a circularly polarized elastic wave in the target isotropic solid. We show that the perfect conversion is possible if the anisotropic medium has effective material properties to exhibit a unique coupled resonance phenomenon at a target frequency; the resonance refers to the simultaneous occurrence of the Fabry-Pérot resonance in one

[1]Department of Mechanical Engineering, Seoul National University, Seoul, South Korea. [2]Department of Applied Nano Mechanics, Korea Institute of Machinery and Materials, Daejeon, South Korea. [3]Institute of Advanced Machines and Design, Seoul National University, Seoul, South Korea. [4]Intelligent Wave Engineering Team, Korea Research Institute of Standards and Science, Daejeon, South Korea. [5]Department of Precision Measurement, University of Science and Technology, Daejeon, South Korea. ✉e-mail: jmkweun@kimm.re.kr; yykim@snu.ac.kr

diagonal plane and the quarter-wave resonance in another diagonal plane. It has been well known that a linearly polarized elastic wave can be fully transmitted to the target medium using either Fabry-Pérot resonance[13] or quarter-wave resonance with impedance matching[14]. However, these two resonances are fundamentally different from the proposed coupled resonance in that either of the two resonances can realize the full transmission of any linearly polarized elastic wave but not the circularly polarized elastic wave. The coupled resonance for generating circularly polarized elastic waves induces these two resonances simultaneously within the anisotropic medium, resulting in the full transmission of two orthogonal linearly polarized elastic waves at the exact phase difference of 90°. We establish the theoretical conditions for the coupled Fabry-Pérot and quarter-wave resonances, which can be turned into the conditions for the effective material properties that the anisotropic medium must satisfy. Using the coupled resonance, it is possible to achieve perfect linear-to-circular polarization conversion without any theoretical limit, overcoming the intrinsic limitations of the classical birefringent plate.

In this paper, the concept of elastic metamaterials[15,16] is utilized to implement the proposed coupled resonance within an anisotropic medium. 3D micro-structuring of elastic metamaterials has provided overwhelming physical properties beyond common homogeneous materials, such as strong ultralight behavior[17,18], auxetic behavior[19,20], nonlinearity[21,22], mechanical chirality[23,24], and fluid-like behavior[25,26]. We propose an obliquely perforated microstructure of a 3D anisotropic elastic metamaterial to simultaneously exploit two distinct resonances. Numerical analysis verifies that the coupled resonance theory is realizable with the proposed 3D anisotropic elastic metamaterial. We fabricated an aluminum-based metamaterial composed of hundreds of high-aspect-ratio holes for experimental validation. We observed circularly polarized ultrasonic shear waves at one end surface of an aluminum-metamaterial-aluminum sandwich system under the linearly polarized shear wave incidence at the other end surface.

Ultrasonic shear waves have been widely used for structural health monitoring and nondestructive testing[27]. Traditionally, the generation of shear waves has been limited to linear polarization along specific directions. This limitation presents a notable drawback: such waves are ineffective in detecting cracks aligned parallel to their polarization direction. Building on our findings presented in this study, we will show that the use of circularly polarized shear waves can identify cracks that remain undetected by their linearly polarized counterparts.

## Results

### Coupled resonance theory

When a linearly polarized elastic wave propagates in the $x$-direction through a 3D isotropic solid, waves in which the medium vibrates in the $x$-, $y$-, and $z$-directions are referred to as longitudinal ("L"), shear horizontal ("SH"), and shear vertical ("SV") waves, respectively. A circularly polarized elastic wave is a combination of two orthogonal linearly polarized elastic waves that have the same magnitude and propagation speed but a 90° phase difference. In an isotropic solid, however, longitudinal waves always propagate faster than shear waves. Consequently, elastic circular polarization can be formed by only two shear waves, SV and SH waves, propagating at the same speed. Our idea is to realize a perfect conversion between linear and circular polarization of shear waves, as well as the decoupling between longitudinal and shear waves in a target isotropic solid (mass density of $\rho_0$ and stiffness of $\mathbf{C_0}$) by introducing a 3D anisotropic elastic metamaterial (effective mass density of $\rho$, effective stiffness of $\mathbf{C}$, and thickness of $d$), as depicted in Fig. 1a.

In this section, we derive the theoretical conditions that the physical properties of the anisotropic metamaterial should satisfy for perfect linear-to-circular polarization conversion. The incident wave can be either an SV or SH wave, and the transmitted wave can be either a left-handed circularly polarized shear ("LCS") or a right-handed circularly polarized shear ("RCS") wave. In Fig. 1, without loss of

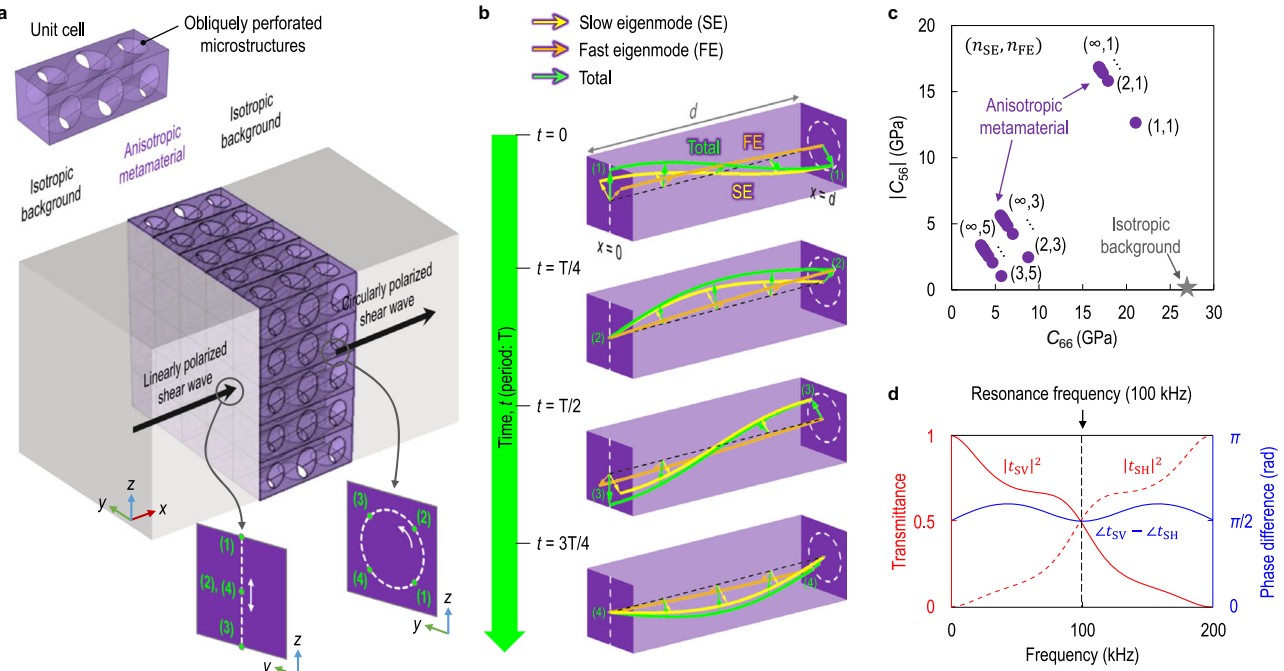

**Fig. 1 | Coupled resonance theory. a** Schematic diagram of a 3D anisotropic elastic metamaterial sandwiched by an isotropic background to realize the perfect polarization conversion from linear to circular. The anisotropic metamaterial is composed of obliquely perforated microstructures. **b** Schematic diagram of the coupled resonance theory. The total displacement field (green) inside the anisotropic metamaterial is decomposed into the displacement fields of the half-wavelength-matched slow eigenmode (yellow) and quarter-wavelength-matched fast eigenmode (orange). **c** Physical properties of the anisotropic metamaterial satisfying the derived theoretical conditions with respect to $n_{SE}$ and $n_{FE}$. The isotropic background is aluminum. **d** Frequency responses of the transmission coefficients under the SV wave incidence to the anisotropic metamaterial satisfying the coupled resonance theory with $n_{SE} = n_{FE} = 1$.

generality, we consider the case of SV-to-LCS polarization conversion. Inside the anisotropic metamaterial, the stress field ($[\sigma_{xx}\,\sigma_{xz}\,\sigma_{xy}]^T$) and the strain field ($[\varepsilon_{xx}\,\gamma_{xz}\,\gamma_{xy}]^T$) are related as

$$
\begin{bmatrix} \sigma_{xx} \\ \sigma_{xz} \\ \sigma_{xy} \end{bmatrix} = \begin{bmatrix} C_{11} & C_{15} & C_{16} \\ & C_{55} & C_{56} \\ \text{sym.} & & C_{66} \end{bmatrix} \begin{bmatrix} \varepsilon_{xx} \\ \gamma_{xz} \\ \gamma_{xy} \end{bmatrix}, \tag{1}
$$

where $C_{11}$, $C_{55}$, and $C_{66}$ correspond to the stiffness constants of L, SV, and SH waves, respectively, while $C_{15}$, $C_{16}$, and $C_{56}$ correspond to the mode-coupling stiffness constants between L-SV, L-SH, and SV-SH waves, respectively. All mode-coupling stiffness constants are always zero in isotropic solids, but they can be non-zero in anisotropic metamaterials.

At the incident interface of the metamaterial, a linearly polarized shear wave splits into two shear eigenmodes formed inside the metamaterial, as shown in Fig. 1b, which play a crucial role in the generation of circularly polarized shear waves. For the analysis, we defined a slow one as a slow eigenmode ("SE") and a fast one as a fast eigenmode ("FE") among two shear eigenmodes. Remind that a circularly polarized shear wave is the sum of two linearly polarized shear waves of equal magnitude but 90° out-of-phase. Thus, the slow and fast eigenmodes should satisfy the following three conditions for perfect linear-to-circular polarization conversion: condition (1) is that the slow and fast eigenmodes should not be coupled with the longitudinal eigenmode; condition (2) is that the slow and fast eigenmodes should be polarized with the same magnitude; condition (3) is that the slow and fast eigenmodes should be fully transmitted through the metamaterial with a 90° phase difference.

In more detail, condition (1) is required to suppress the transmission and reflection of longitudinal waves under the shear wave incidence, and it can be satisfied by zero longitudinal-shear mode-coupling stiffness constants of the metamaterial (i.e., $C_{15}=C_{16}=0$). Condition (2) is necessary to accomplish the magnitude balance between transmitted SV and SH waves in the isotropic background. To this end, the slow and fast eigenmodes should be polarized in two diagonal directions in the $y$–$z$ plane. If the eigenmodes are not diagonally polarized in the $y$–$z$ plane, the magnitudes of slow and fast eigenmodes are determined differently, leading to distorted circular polarization (see Supplementary Fig. 1). For diagonal polarization inside the metamaterial, its shear stiffness constants of SV and SH waves should be equal (i.e., $C_{55}=C_{66}$). The detailed derivation is described in Supplementary Note 2. The intermediate conclusion so far is that conditions (1) and (2) can be achieved with $C_{15}=C_{16}=0$ and $C_{55}=C_{66}$, respectively.

Condition (3) is required to achieve the 90° out-of-phase relation between transmitted SV and SH waves in the isotropic background with the full (100%) energy efficiency of the polarization conversion. In order to accomplish this, our idea is comprehensively employing two distinct resonances for full transmission: Fabry-Pérot resonance and quarter-wave resonance with impedance matching. Fabry-Pérot and quarter-wave resonances are characterized in that the thickness of the metamaterial must be matched by the half-wavelength and quarter-wavelength of the wave propagating through the metamaterial. When the slow and fast eigenmodes inside the metamaterial are matched by half-wavelength (i.e., $d = n_{SE} \cdot \frac{\lambda_{SE}}{2}$ ($n_{SE}=1,2,3\ldots$)) and quarter-wavelength (i.e., $d = n_{FE} \cdot \frac{\lambda_{FE}}{4}$ ($n_{FE}=1,3,5\ldots$)), their 90° phase difference is always guaranteed because $\Delta\phi_{SE} = \frac{2\pi}{\lambda_{SE}} \cdot d = n_{SE} \cdot \pi$ and $\Delta\phi_{FE} = \frac{2\pi}{\lambda_{FE}} \cdot d = \frac{n_{FE}\cdot\pi}{2}$. Here, $\Delta\phi_{SE}$ and $\Delta\phi_{FE}$ denote the phase changes of the slow and fast eigenmodes through the metamaterial, and $\lambda_{SE}$ and $\lambda_{FE}$ denote the wavelengths of the slow and fast eigenmodes inside the metamaterial. Importantly, the wavelengths of the slow and fast eigenmodes can be modulated by the SV-SH mode-coupling stiffness constant of the

metamaterial as follows (see Supplementary Note 2 for derivation):

$$
\lambda_{SE} = \frac{1}{f} \cdot \sqrt{\frac{C_{66} - |C_{56}|}{\rho}}, \quad \lambda_{FE} = \frac{1}{f} \cdot \sqrt{\frac{C_{66} + |C_{56}|}{\rho}}, \tag{2}
$$

where $f$ indicates a frequency. Most importantly, we could explicitly determine the physical properties ($\rho$ and $\mathbf{C}$) of the anisotropic metamaterial from conditions (1), (2), and (3) as follows (see Supplementary Note 3 for derivation):

$$
\rho = \frac{1}{4f \cdot d} \cdot n_{FE} \cdot \sqrt{\rho_0 C_{66}^0}, \tag{3}
$$

$$
C_{15} = C_{16} = 0, \tag{4}
$$

$$
C_{55} = C_{66} = \frac{1}{2} f \cdot d \cdot n_{FE} \cdot \left( \frac{4}{n_{FE}^2} + \frac{1}{n_{SE}^2} \right) \cdot \sqrt{\rho_0 C_{66}^0}, \tag{5}
$$

$$
|C_{56}| = \frac{1}{2} f \cdot d \cdot n_{FE} \cdot \left( \frac{4}{n_{FE}^2} - \frac{1}{n_{SE}^2} \right) \cdot \sqrt{\rho_0 C_{66}^0}. \tag{6}
$$

Here, we note that $C_{11}$ does not affect the transmission and reflection of shear waves because of Eq. (4). The spatiotemporal displacement fields in Fig. 1b support that the linearly polarized input displacement vector is perfectly converted into the circularly polarized output displacement vector as an elastic wave propagates through the metamaterial that satisfies the theoretical conditions in Eqs. (3)–(6). A video demonstration of Fig. 1b can be found in Supplementary Movie 1.

Multiple sets of physical properties of anisotropic metamaterials for the perfect polarization conversion in isotropic aluminum ($\rho_0 = 2700\ \text{kg m}^{-3}$, $C_{55}^0 = C_{66}^0 = 26.3\ \text{GPa}$, and $C_{56}^0 = 0\ \text{GPa}$) are depicted in Fig. 1c. The target resonance frequency and metamaterial thickness were chosen as $f = 100\ \text{kHz}$ and $d = 0.01\ \text{m}$, respectively. In Fig. 1d, we calculate the transmission spectra of the metamaterial with $n_{SE}=n_{FE}=1$ where $t_{SV}$ and $t_{SH}$ denote transmission coefficients of SV and SH waves. The result demonstrates that the magnitude balance ($|t_{SV}|^2 = |t_{SH}|^2 = 0.5$) with the 90° out-of-phase relation ($|\angle t_{SV} - \angle t_{SH}| = 0.5\pi$) is achieved at the resonance frequency of 100 kHz, resulting in the generation of exact circular polarization in isotropic aluminum with full energy efficiency. See Supplementary Note 4 and Supplementary Figs. 2 and 3 for transmission and reflection responses of the metamaterial over the broader frequency range of 0 to 400 kHz.

## Metamaterial realization

Designing microstructures of metamaterials that exhibit desired effective physical properties is challenging yet crucial. In this paper, we utilize unit cells shaped as rectangular parallelepipeds. These cells feature obliquely perforated cylindrical holes of varying sizes. This design is implemented to achieve the specific anisotropy described in Eqs. (3)–(6). The geometry of the unit cell is characterized by seven parameters, $L_x, L_y, L_z, r_1, r_2, r_3$, and $L$, as shown in Fig. 2a. Symbols $L_x$, $L_y$, and $L_z$ denote the unit cell dimensions in the $x$-, $y$-, and $z$-directions, respectively. The radii of the circular cross-sections of three different cylindrical holes are denoted by $r_1$, $r_2$, and $r_3$. The holes are arranged at an interval of $L$ in the $x$-direction, which is the wave propagation direction. Here, we note that the cylindrical holes are drilled in the direction perpendicular to the $x$-axis for the realization of $C_{15}=C_{16}=0$. In addition, the cylindrical holes are rotated by 45°, allowing the metamaterial to attain $C_{55}=C_{66}$. Most notably, the metamaterial can exhibit a non-zero $C_{56}$ because the drilling direction is not parallel to the $y$- or $z$-axes. The $C_{56}$ value can be tailored to satisfy the coupled resonance theory by adjusting the radii of the cylindrical holes constituting the

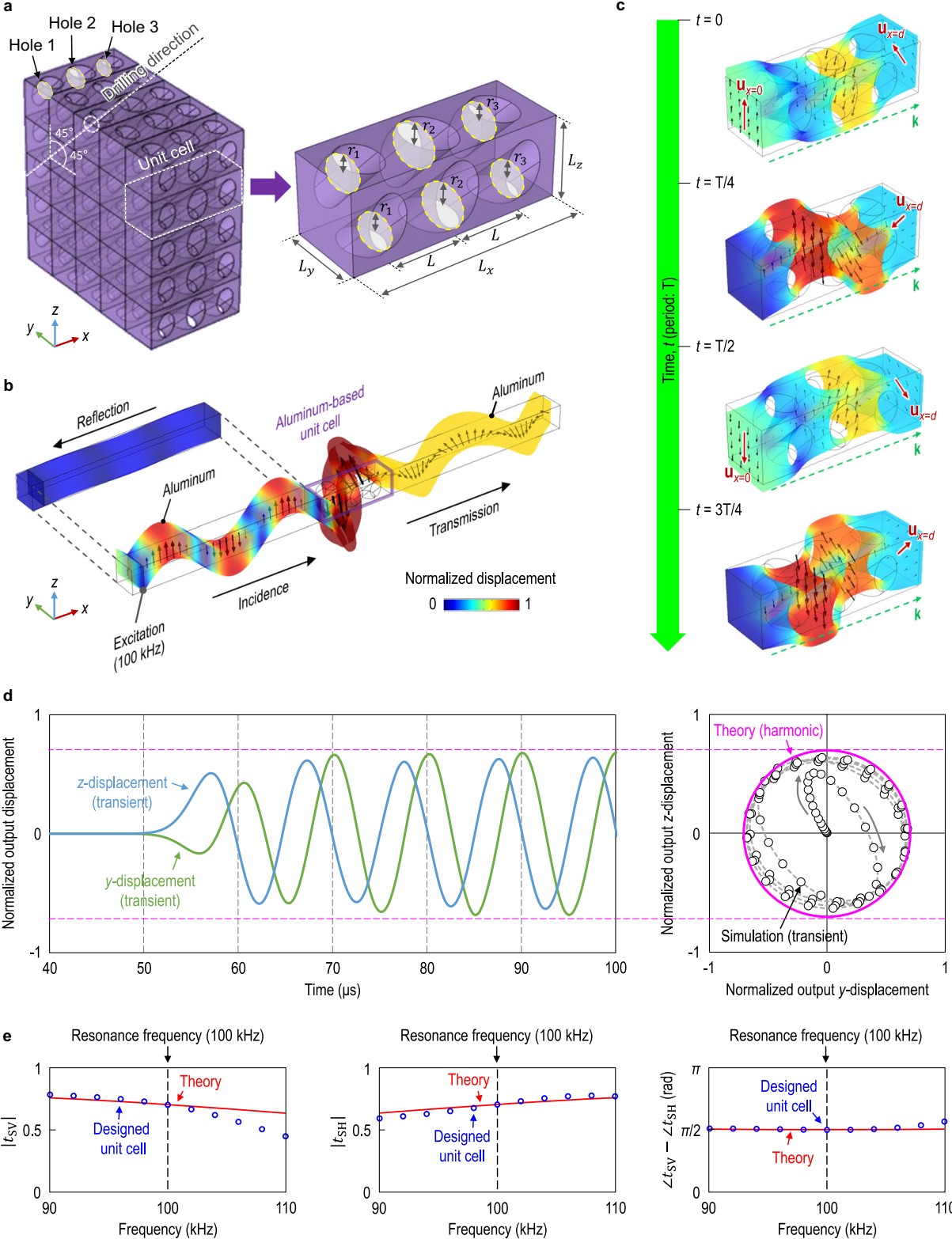

**Fig. 2 | Realization with 3D anisotropic metamaterials. a** The unit cell of the 3D anisotropic metamaterial with three different-sized holes. **b** The time-harmonic numerical simulation result with the designed unit cell. The unit cell is made of aluminum and inserted between aluminum (periodic conditions in the *y*- and *z*-directions). The SV wave is incident on the unit cell in the *x*-direction at 100 kHz. **c** The spatiotemporal displacement field inside the designed unit cell at the resonance frequency. $\mathbf{u}_{x=0}$ and $\mathbf{u}_{x=d}$ represent the displacement vector at the incident and transmission interfaces of the designed unit cell, respectively, and **k** represents the wave propagation direction. **d** The time-transient numerical simulation result with the designed unit cell. The output *y*- and *z*-displacements through the metamaterial are presented with the theoretical steady-state displacement trajectory (magenta) assuming full transmission. **e** Simulated transmission spectra of the designed unit cell (blue) with the theoretical transmission spectra (red) exhibiting the coupled resonance with $n_{SE} = 2$ and $n_{FE} = 3$. The magnitudes of the transmitted SV and SH waves are presented on the left and middle, and their phase difference is presented on the right.

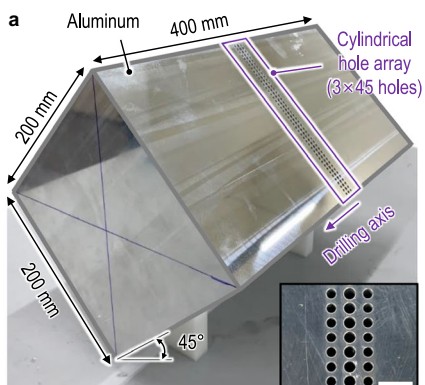
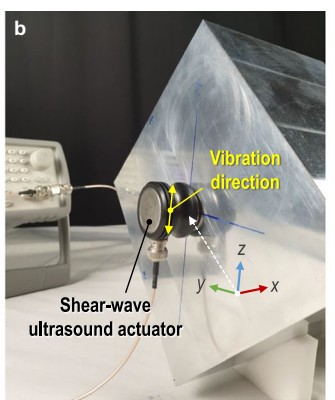
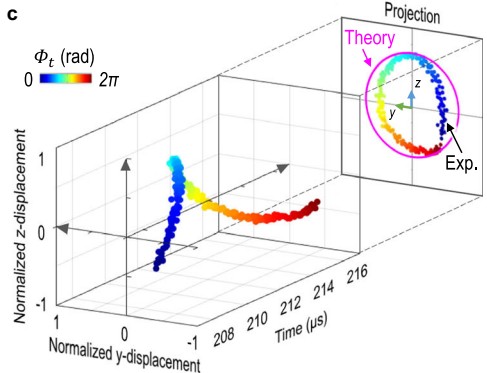

**Fig. 3 | Experimental validation. a**, The fabricated aluminum-metamaterial-aluminum sandwich system. The bottom right inset represents a zoomed-in cylindrical hole array (the scale bar is 10 mm). **b**, The installation of the shear-wave ultrasound actuator. **c**, The measured time-transient output shear-wave displacement through the metamaterial (normalized to the magnitude of the input shear-wave displacement). The measured data is indicated by dots that change color from blue to red over time phase $\phi_t$, while the theoretical steady-state displacement trajectory assuming full transmission is indicated by a magenta circle.

metamaterial. For the realization of perfect elastic circular polarization in isotropic solids, the geometric parameters of the unit cell were optimized using a gradient-based algorithm[28]. In practice, an aluminum-based anisotropic metamaterial was designed to achieve the perfect SV-to-LCS polarization conversion in aluminum at 100 kHz. The optimization result of the geometric parameters is given in Supplementary Table 1. See Supplementary Note 5 and Supplementary Figs. 4 and 5 for the realization of other types of linear-to-circular polarization conversion (SV-to-RCS, SH-to-LCS, and SH-to-RCS).

For numerical validation of the proposed metamaterial, finite element analysis-based simulations were conducted. The time-harmonic simulation result in Fig. 2b confirms that the metamaterial perfectly converts the incident SV wave into the LCS wave at the resonance frequency of 100 kHz. Quantitatively, the metamaterial produced highly accurate circular polarization with $\frac{|t_{SH}|}{|t_{SV}|} = 1.004$ and $\frac{|\angle t_{SV} - \angle t_{SH}|}{0.5\pi} = 0.998$, which should both be 1 for circular polarization to be exact. Moreover, nearly zero reflection can be found in Fig. 2b, indicating the nearly full transmission through the metamaterial. Indeed, the transmission and reflection coefficients of longitudinal waves were identically zero. The spatiotemporal displacement fields inside the proposed unit cell, presented in Fig. 2c, also support that incident linear polarization is converted into circular polarization as the wave passes through the metamaterial. A video demonstration of Fig. 2c is provided in Supplementary Movie 2. In addition, time-transient simulations were conducted to demonstrate the non-steady-state response of the proposed metamaterial, and the result in Fig. 2d validates that the transmitted LCS wave can be formed within three sinusoidal cycles of the incident SV wave. Moreover, the simulated time-transient circular particle motion was nearly identical to the theoretically calculated time-harmonic circular trajectory under the assumption of full transmission. A video demonstration of the time-transient simulation can be found in Supplementary Movie 3.

We evaluated the effective physical properties of the designed 3D anisotropic metamaterial to confirm that the metamaterial is compatible with the proposed coupled resonance theory. To this end, we extended the scattering parameter retrieval method, which was established for 2D elastic metamaterials[29–31], to 3D elastic metamaterials (see Methods for details). The calculated effective density and stiffness constants of the designed metamaterial are $\rho = 3024 \text{ kg m}^{-3}$, $C_{55} = C_{66} = 16.6 \text{ GPa}$, and $C_{56} = -4.7 \text{ GPa}$ at the frequency of 100 kHz. Through the effective physical properties of the metamaterial, the phase changes of the slow and fast eigenmodes through the metamaterial were estimated as $\Delta\phi_{SE} = 2.004\pi \approx 2 \times \pi$ and $\Delta\phi_{FE} = 1.499\pi \approx 3 \times 0.5\pi$. In other words, the metamaterial fulfills the coupled resonance theory with $n_{SE} = 2$ and $n_{FE} = 3$, which eventually leads to perfect

polarization conversion from linear to circular in aluminum. Figure 2e shows that the simulated transmission spectra of the metamaterial correspond well with the theoretical transmission spectra that exhibit the coupled resonance with $n_{SE} = 2$ and $n_{FE} = 3$ at 100 kHz.

The proposed single-phase microstructure in Fig. 2a has the advantage of being easy to fabricate while effectively realizing a non-zero $C_{56}$. However, it is also possible to design metamaterial microstructures with different geometries or materials (see Supplementary Fig. 6 for other microstructure candidates). In addition, we choose aluminum as the target isotropic background material and 100 kHz as the target resonance frequency in Fig. 2b–e. However, the proposed coupled resonance is applicable to other materials and frequencies (see Supplementary Note 6 for the applicability and limitation of the proposed method). Supplementary Table 2 provides the required physical properties of the anisotropic medium to realize the coupled resonance at various target materials and frequencies. The geometric parameters of the metamaterial microstructure with another geometric shape or at another target frequency can be obtained by utilizing the same shape optimization algorithm[28].

## Experimental validation

To experimentally verify that the coupled resonance theory can generate perfect elastic circular polarization within isotropic solids, we fabricated the aluminum-based anisotropic metamaterial designed in Fig. 2 and conducted ultrasound experiments. Specifically, a super-drilling procedure was used to machine the $3 \times 45$ high-aspect-ratio cylindrical hole array in the center of the large-scale bulk aluminum specimen. Then, the specimen was rotated 45° about the $x$-axis, as shown in Fig. 3a, to allow the wave to pass through the obliquely perforated microstructures. Shear-wave ultrasound actuators, as shown in Fig. 3b, were utilized as the shear-wave transmitter and receiver. The incident wave was an SV wave with a 5-cycle sinusoidal waveform and a frequency of 100 kHz. To visualize the transmitted elastic circular polarization through the metamaterial, the temporal trajectory of the measured output shear-wave displacement was presented in Fig. 3c. The result shows that the circularly rotating trajectory was observed, and the radius of the measured trajectory was close to the radius of the theoretically calculated steady-state circular trajectory assuming full transmission. The differences between the theory and the experiment could be induced by the distorted plane-wave generation due to the finite size of the wave source and unavoidable wave reflections at the boundary of the specimen. Nevertheless, our experiments demonstrated that nearly perfect circular polarization of elastic waves in aluminum, a common and widely used isotropic solid,

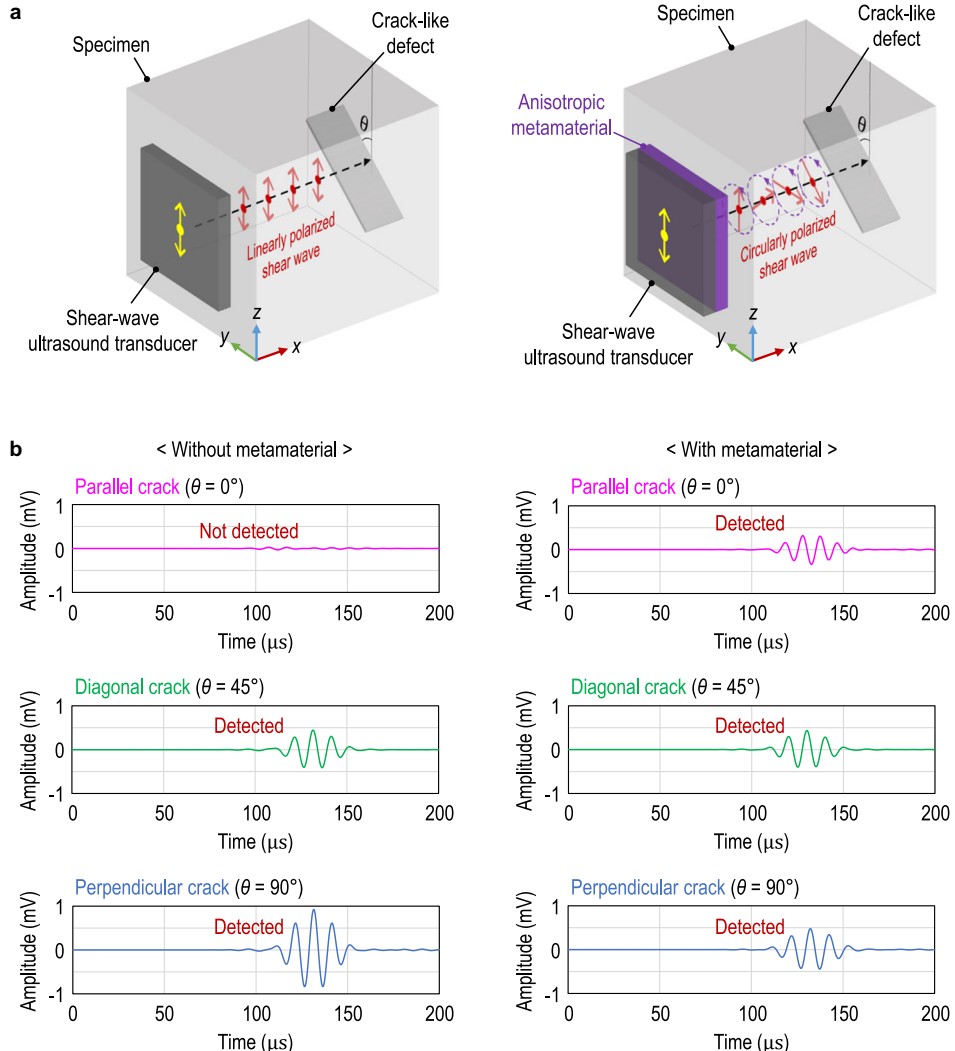

**Fig. 4 | Non-destructive testing using elastic circular polarization. a** An illustration of crack-like defect detection using linearly polarized (left) and circularly polarized (right) shear waves. $\theta$ represents the orientation of the crack. **b** The simulated reflected signals from parallel, diagonal, and perpendicular cracks without (left) and with (right) the proposed metamaterial. The specimen is aluminum, and the metamaterial designed in Fig. 1d is used at 100 kHz.

could be realized with the anisotropic metamaterial based on the coupled resonance theory.

## Arbitrarily oriented crack detection

As an unprecedented application of a circularly polarized shear wave in non-destructive testing, we suggest the detection of arbitrarily oriented cracks, which are difficult to detect using a linearly polarized shear wave alone. In ultrasonic non-destructive testing, the crack is detected by receiving the reflected ultrasonic shear wave from the crack. Nonetheless, it is impossible to detect the crack using a linearly polarized shear wave if the polarization direction and the orientation of the crack are parallel[32]. To overcome this limitation in non-destructive testing, we propose employing a circularly polarized shear wave, as illustrated in Fig. 4a. Here, we used finite element analysis to simulate ultrasonic testing. Notably, a circularly polarized shear wave can interact with cracks of *any* orientation because its polarization direction rotates in space and time. For comparison, reflected signals from parallel, diagonal, and perpendicular cracks are simulated with and without the use of circular polarization, as shown in Fig. 4b. The signal reflected from the crack was extracted as the difference between the reflected signals with and without the crack. Without the use of a circularly polarized shear wave, the presence of the parallel

crack cannot be identified because there is no reflected signal from the parallel crack. However, a circularly polarized shear wave can distinguish all the reflected signals from parallel, diagonal, and perpendicular cracks. Cracks of any orientation can now be inspected at once using circular polarization, allowing for faster and more accurate ultrasonic non-destructive testing.

## Discussion

The following conclusions can be made as we investigated the so-far-unexplored circular polarization of elastic waves in common isotropic solids. We discovered that it is possible to perfectly generate a circularly polarized elastic wave by passing a linearly polarized elastic wave through an anisotropic elastic medium having specific physical properties. We derived the theoretical conditions required for the linear-to-circular polarization-converting medium realized by a metamaterial. In addition, the physics underpinning the perfect polarization conversion, the coupled resonance between Fabry-Pérot and quarter-wave resonances, was revealed. Our theory was confirmed through the ultrasonic wave experiment.

Unlike the lack of exploration in circularly polarized elastic waves, circularly polarized light has been intensively explored and fertilized optical techniques that linearly polarized light cannot do, including

3D projection[33], high-performance display[34,35], high-resolution molecular structure analysis[36–38], and quantum computing[39,40]. Accordingly, our work is expected to open a new horizon for the elastic version of all circular-polarization-based optical devices. As an example of a potential application, the detection of an arbitrarily oriented crack using circularly polarized ultrasonic shear waves, which is otherwise difficult to realize, was demonstrated.

Noting that the spinning nature of elastic waves has recently received considerable attention[41–48], our findings can be a promising strategy to efficiently generate various elastic spins. For instance, it may be possible to generate a circularly polarized bulk shear wave from a linearly polarized bulk longitudinal wave by comprehensively utilizing the longitudinal-to-shear mode conversion and the linear-to-circular polarization conversion. Furthermore, in a special 3D solid medium with the same longitudinal and shear stiffness, a novel circularly polarized bulk elastic wave composed of longitudinal and shear waves can be observed.

## Methods
### Theoretical details
In Fig. 1, to calculate the scattering parameters of a 3D anisotropic metamaterial sandwiched by an isotropic background medium, we construct the $6 \times 6$ scattering matrix for the case of the SV wave incidence as

$$\begin{bmatrix} t_L e^{i\phi_L} \\ 0 \\ t_{SV} e^{i\phi_S} \\ 0 \\ t_{SH} e^{i\phi_S} \\ 0 \end{bmatrix} = \begin{bmatrix} S_{11} & S_{12} & S_{13} & S_{14} & S_{15} & S_{16} \\ & S_{22} & S_{23} & S_{24} & S_{25} & S_{26} \\ & & S_{33} & S_{34} & S_{35} & S_{36} \\ & & & S_{44} & S_{45} & S_{46} \\ & \text{sym.} & & & S_{55} & S_{56} \\ & & & & & S_{66} \end{bmatrix} \begin{bmatrix} 0 \\ r_L \\ 1 \\ r_{SV} \\ 0 \\ r_{SH} \end{bmatrix}, \quad (7)$$

where $\phi_L$ and $\phi_S$ are the phase correction of longitudinal and shear waves for wave propagation through the metamaterial, respectively. Here, we define the transmission (reflection) coefficients of L, SV, and SH waves as $t_L$, $t_{SV}$ and $t_{SH}$ ($r_L$, $r_{SV}$ and $r_{SH}$). For the generation of exact circular polarization, the transmission coefficients of two shear waves should satisfy $|t_{SV}| = |t_{SH}|$ and $|\angle t_{SV} - \angle t_{SH}| = 0.5\pi$. For full energy efficiency of the polarization conversion, all other transmission and reflection coefficients except $t_{SV}$ and $t_{SH}$ should be zero, i.e., $t_L = r_L = r_{SV} = r_{SH} = 0$. The scattering matrix is determined by the physical properties of the anisotropic metamaterial[29–31]. In Fig. 1d, we used the physical properties of the metamaterials as $\rho = 2107 \text{ kg m}^{-3}$, $C_{55} = C_{66} = 21.1 \text{ GPa}$, and $C_{56} = -12.6 \text{ GPa}$.

As a homogenization method of the designed 3D anisotropic metamaterial, we find effective physical properties with the same magnitude and phase of scattering parameters as those calculated from the metamaterial[29]. The entire metamaterial is treated as having homogenized effective physical properties[30,31]. Transmission coefficients of the designed metamaterial were numerically calculated as $t_L = 0$, $t_{SV} = 0.0527 + 0.7016i$ and $t_{SH} = -0.6988 + 0.0551i$. Reflection coefficients were calculated as $r_L = 0$, $r_{SV} = 0.0180 + 0.0038i$, and $r_{SH} = -0.1023 - 0.0519i$. The retrieved effective density and stiffness constants of the designed metamaterial are $\rho = 3024 \text{ kg m}^{-3}$, $C_{55} = C_{66} = 16.6 \text{ GPa}$, and $C_{56} = -4.7 \text{ GPa}$. Indeed, the effective physical properties of the metamaterial satisfy the theoretical values (exhibiting the coupled resonance with $n_{SE} = 2$ and $n_{FE} = 3$) obtained by Eqs. (3)–(6) within the numerical error of 5%.

### Numerical details
Commercial software (COMSOL Multiphysics) based on finite element analysis was used for the numerical analysis. For the physical properties of aluminum, the mass density of 2700 kg m⁻³, Young's modulus of 70 GPa, and Poisson's ratio of 0.33 were used. In Fig. 2b,

periodic conditions were applied in the $y$- and $z$-directions while waves propagated in the $x$-direction, and low-reflecting boundaries were applied on the right and left ends of the simulation model. Color surfaces in Fig. 2b, c represent normalized displacements (normalized to the displacement magnitude of the incident wave), and black arrows represent total displacement vectors. In Fig. 4, the specimen is an aluminum cube with a side length of 150 mm. The crack-like defect is a cavity with a size of $30 \times 3 \times 100 \text{ mm}^3$. The size of the transducer is $10 \times 100 \times 100 \text{ mm}^3$. The size of the metamaterial is also $10 \times 100 \times 100 \text{ mm}^3$. The shear-wave ultrasound transducer generates an enveloped 5-cycle sinusoidal SV wave with a center frequency of 100 kHz.

### Experimental details
The width, height, and length of the fabricated aluminum-metamaterial-aluminum sandwich system are 200 mm, 200 mm, and 400 mm. The cylindrical holes were directly machined by a super-drilling process with additional wire cutting (Mitsubishi DWC-300HA) and milling machining (DOOSAN MYNX 6500). Three different-sized holes were arranged at an interval of 5.43 mm along the $x$-axis parallel to the wave propagation direction, and forty-five hole arrays were arranged at an interval of 4.36 mm in the diagonal direction along the positive $y$- and positive $z$-axis. Considering the hole-size limit of the fabrication, the radii of the cylindrical holes of 1.43 mm, 1.73 mm, and 1.47 mm were determined.

Shear-wave ultrasound actuators (Olympus® V1548) with a diameter of 25.4 mm were bonded on the left and right sides of the specimen by shear-wave couplant (Olympus® SWC-2). An SV wave with a 5-cycle sinusoidal waveform was generated by the function generator (Agilent Technologies 33220 A). The transmitted SV and SH waves through the metamaterial were recorded by the digital oscilloscope (LeCroy WaveRunner™ 620Zi). The signal transmitted through the metamaterial in Fig. 3c was normalized to the signal transmitted through the reference aluminum specimen ($200 \times 200 \times 400 \text{ mm}^3$) without the metamaterial. The average value of a total of five measurements was used.

## Data availability
Data that support the findings of this study are available from the corresponding authors upon request.

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

## Acknowledgements

This research was supported by the Global Frontier R&D Program on Center for Wave Energy Control based on Metamaterials (CAMM-2014M3A6B3063711) contracted through the Institute of Advanced Machines and Design at Seoul National University funded by the Korea Ministry of Science, ICT & Future Planning. This research was also supported by the National Research Foundation of Korea (NRF-2022R1A2C2008067) funded by the Korea Ministry of Science, ICT & Future Planning. J.L. acknowledges the Global PhD Fellowship (NRF-2019H1A2A1075829) funded by the Korean Ministry of Education.

## Author contributions

Y.Y.K. supervised the project. J.L. and M.J.K. conceived ideas for the project and performed the theoretical analysis. J.L. performed numerical simulations. J.L., M.J.K., W.L., and H.M.S. performed the experiments. J.L. and Y.Y.K. wrote the manuscript.

## Competing interests

The authors declare no competing interests.
