## [Peer Review File · Nature Communications]

REVIEWER COMMENTS

Reviewer #1 (Remarks to the Author):

The manuscript describes a novel approach for the transformation of linearly polarized elastic waves into circularly polarized counterparts, leveraging the unique properties of anisotropic metamaterials. A comprehensive theoretical examination is conducted, yielding essential prerequisites for achieving the conversion of linear to circular polarization within the medium. Moreover, the underlying physical principles governing flawless polarization conversion are unveiled. Empirical validation of the proposed methodology is carried out through experimentation. Furthermore, the practical relevance of this method is explained by presenting an illustrative application involving the utilization of circularly polarized ultrasonic shear waves for the detection of cracks in arbitrary orientations. I believe that the topic of this manuscript is very interesting and innovative, which will notably contribute to better understanding and application of the circularly polarized waves. Altogether I would recommend the work for publication with some details to be addressed.

1. In line 203 and 335, the value of density is $\rho=3024\text{kgm}^{-3}$ and $\rho=2700\text{kgm}^{-3}$, but the value of density is $\rho=3,024\text{kgm}^{-3}$ in line 322 and $\rho=2,700\text{kg}\cdot\text{m}^{-3}$ in line 144. I think it is necessary to unify numeric expressions.
2. The formula S1 is missing a symbol in line 29 in the Supplementary Note 1.
3. The subgraph in Fig. 2(a) would better be enlarged and clearly marked with how the holes rotated, so that the details can be better seen and understood.
4. It is suggested to provide some specific potential application cases in the introduction to more clearly demonstrate the practical significance and potential of this research.
5. It is suggested to further discuss the applicability and limitations of the proposed method. This includes applicability to different materials, frequency ranges, and geometric shapes.
6. How to realize different target frequencies (50kHz, 5kHz, 500Hz...or lower)? Why the authors chose obliquely perforated microstructures to design the special metamaterials, whether microstructures with different parts or materials can be used to design such metamaterials.

Reviewer #2 (Remarks to the Author):

This study summarizes theoretical and experimental results of an ingenious mechanism for utilizing a metamaterial with appropriate bulk elastic parameters, coupled with an anisotropic Fabry-Perot resonance cell, to generate circularly polarized elastic waves at ultrasonic frequencies (e.g, 100 kHz). The presentation is well-articulated and the results are appropriately demonstrated and supported experimentally. Although interest in generating circularly polarized transverse elastic wave fields is long-standing (e.g., Einspruch, 1964), this methodology appears to me to be a significant advance in this area and has good potential to advance ultrasonic technology for material testing and other purposes. I did not find any errors in the theoretical or experimental presentation and believe that the description of

the apparatus and metamaterial design (i.e., drilled aluminum) is sufficient so that the results could be reproduced and extended.

A few minor comments -- 1) The study gives short-shrift to the considerable work in the seismological community regarding shear-wave splitting at the mantle and crustal scale. In these circumstances the birefringence of Earth materials due to bulk mineral alignment and/or stress-related cracks creates transient elliptical or circular polarizations that have been studied for decades. 2) The accompanying supplemental materials were apparently incomplete in the sense that the video demonstration/supplemental movies were only (for me) downloadable as a static Powerpoint -- I was looking forward to seeing these.

Point-by-Point Response to the Reviewers' Comments

Response to the Reviewer #1

General Comment: The manuscript describes a novel approach for the transformation of linearly polarized elastic waves into circularly polarized counterparts, leveraging the unique properties of anisotropic metamaterials. A comprehensive theoretical examination is conducted, yielding essential prerequisites for achieving the conversion of linear to circular polarization within the medium. Moreover, the underlying physical principles governing flawless polarization conversion are unveiled. Empirical validation of the proposed methodology is carried out through experimentation. Furthermore, the practical relevance of this method is explained by presenting an illustrative application involving the utilization of circularly polarized ultrasonic shear waves for the detection of cracks in arbitrary orientations. I believe that the topic of this manuscript is very interesting and innovative, which will notably contribute to better understanding and application of the circularly polarized waves. Altogether I would recommend the work for publication with some details to be addressed.

General Response: Thank you for supporting the publication of our paper in Nature Communications. Detailed responses to each comment are provided below.

Comment #1: In line 203 and 335, the value of density is $\rho=3024\text{kgm}^{-3}$ and $\rho=2700\text{kgm}^{-3}$, but the value of density is $\rho=3,024\text{kgm}^{-3}$ in line 322 and $\rho=2,700\text{kg}\cdot\text{m}^{-3}$ in line 144. I think it is necessary to unify numeric expressions.

Response: In the revised manuscript, we have unified all numerical expressions as follows:

In the revised manuscript:

(Page 7)

(Page 10)

(Page 18)

(Page 19)

(Page 19) the mass density of

Comment #2: The formula S1 is missing a symbol in line 29 in the Supplementary Note 1.

Response: In the revised supplementary information, we have added the missing symbol as follows:

In the revised supplementary information:

(Page 1)

Comment #3: The subgraph in Fig. 2(a) would better be enlarged and clearly marked with how the holes rotated, so that the details can be better seen and understood.

Response: To reflect the reviewer’s comment, we have modified Fig. 2(a) to better represent the microstructure of the proposed metamaterial. In addition, in the main text, we have revised the sentences explaining Fig. 2(a) as follows:

In the revised manuscript:

(Page 9) “In this paper, we utilize unit cells shaped as rectangular parallelepipeds. These cells feature obliquely perforated cylindrical holes of varying sizes. This design is implemented to achieve the specific anisotropy described in Eq. (3). The geometry of the unit cell is characterized by seven parameters, L_x , L_y , and L_z , as shown in Fig. 2a. Symbols r_1 , r_2 , and r_3 denote the unit cell dimensions in the x -, y -, and z -directions, respectively. The radii of the circular cross-sections of three different cylindrical holes are denoted by r_1 , r_2 , and r_3 . The holes are arranged at an interval of L in the x -direction, which is the wave propagation direction. Here, we note that the cylindrical holes are drilled in the direction perpendicular to the x -axis for the realization of ϵ_{xx} . In addition, the cylindrical holes are rotated by 45° , allowing the metamaterial to attain ϵ_{xy} . Most notably, the metamaterial can exhibit a non-zero ϵ_{yz} because the drilling direction is not parallel to the y - or z -axes. The ϵ_{yz} value can be tailored to satisfy the coupled resonance theory by adjusting the radii of the cylindrical holes constituting the metamaterial.”

Figure 2(a) (revised)

Comment #4: It is suggested to provide some specific potential application cases in the introduction to more clearly demonstrate the practical significance and potential of this research.

Response: To reflect the reviewer’s comment, we have added an explanation in the introduction about how circularly polarized elastic waves can be practically used in ultrasonic non-destructive testing. Thank you for helping us deliver the significance and potential of our research to a broad readership.

In the revised manuscript (newly added):

(Page 3) “Ultrasonic shear waves have been widely used for structural health monitoring and nondestructive testing, as reviewed in [27]. Traditionally, the generation of shear waves has been limited

to linear polarization along specific directions. This limitation presents a notable drawback: such waves are ineffective in detecting cracks aligned parallel to their polarization direction. Building on our findings presented in this study, we will show that the use of circularly polarized shear waves can identify cracks that remain undetected by their linearly polarized counterparts.”

[27] Miao, H. & Li, F. Shear horizontal wave transducers for structural health monitoring and nondestructive testing: A review, *Ultrasonics* **114**, 106355 (2021).

Comment #5: It is suggested to further discuss the applicability and limitations of the proposed method. This includes applicability to different materials, frequency ranges, and geometric shapes.

Response: In the original manuscript, we chose aluminum as the target material and 100 kHz as the target frequency, just as an example. Indeed, the proposed coupled resonance is applicable to other isotropic background materials and frequencies. To show this, we have newly provided the required physical properties of the anisotropic medium to realize the coupled resonance at various target frequencies (50, 100, and 150 kHz) when the target materials are aluminum (metal) and PEEK (plastic).

A limitation of the proposed method is that there may be difficulties in manufacturing metamaterial microstructures depending on the target material and frequency. For example, as the target frequency increases, a smaller subwavelength-scale microstructure is required. However, if the required radius of the cylindrical hole becomes too small, the microstructure may not be manufactured by the super-drilling procedure used in our experiment.

The proposed coupled resonance can be implemented with metamaterial microstructures of different geometric shapes. We will explain this in more detail in our response to Comment #6. Overall, to reflect the reviewer’s comment, we have added the following sentences in the revised manuscript and supplementary information:

In the revised manuscript (newly added):

(Page 11) “In addition, we choose aluminum as the target isotropic background material and 100 kHz as the target resonance frequency in Figs. 2b-2e. However, the proposed coupled resonance is applicable to other materials and frequencies (see Supplementary Note 6 for the applicability and limitation of the proposed method). Supplementary Table 1 provides the required physical properties of the anisotropic medium to realize the coupled resonance at various target materials and frequencies.”

In the revised supplementary information (newly added):

(Page 5) “Supplementary Note 6: Applicability and limitation of the proposed method

In the main text, we chose aluminum as the target isotropic background material and 100 kHz as the target frequency. However, the proposed coupled resonance is applicable to other isotropic background materials and frequencies. Supplementary Table 1 provides the required physical properties of the anisotropic medium to realize the coupled resonance at various target frequencies (50, 100, and 150 kHz) when the target materials are aluminum (metal) and PEEK (plastic). The density of aluminum and PEEK is 2700 kg m^{-3} and 1320 kg m^{-3} , and the shear stiffness of aluminum and PEEK is 26.3 GPa and 1.51 GPa. Without loss of generality, the thickness of the utilized anisotropic medium with is assumed to be .

The effectiveness of the proposed method is contingent on the feasibility of manufacturing metamaterial microstructures, which varies based on the target material and frequency. For instance, higher target frequencies necessitate increasingly smaller subwavelength-scale microstructures. However, current manufacturing technologies, like the drilling procedure employed in our experiments, may not be capable of producing these intricate microstructures.”

Supplementary Table 1 (newly added)

Background: Aluminum	[kg m ³]	[GPa]	[GPa]	[GPa]
50 kHz	4213.4	10.5	10.5	-6.30
100 kHz	2106.7	21.1	21.1	-12.6
150 kHz	1404.5	31.6	31.6	-19.0
Background: PEEK	[kg m ³]	[GPa]	[GPa]	[GPa]
50 kHz	705.9	1.76	1.76	-1.06
100 kHz	353.0	3.53	3.53	-2.12
150 kHz	235.3	5.29	5.29	-3.18

Comment #6: How to realize different target frequencies (50kHz, 5kHz, 500Hz...or lower)? Why the authors chose obliquely perforated microstructures to design the special metamaterials, whether microstructures with different parts or materials can be used to design such metamaterials.

Response: In our response to Comment #5, we have shown that the proposed resonance is applicable to different target frequencies. However, according to Eq. (3) in the text, for very low target frequencies, large effective density and small effective stiffness are simultaneously required if the thickness (d) of the anisotropic medium is fixed. To avoid this issue, the thickness of the anisotropic medium may be reset in inverse proportion to the target frequency.

The reason we chose obliquely perforated microstructures to design the metamaterials is to realize non-zero SV-SH mode coupling stiffness () practically. Because the proposed microstructure is a single phase, it can be easily fabricated through a super-drilling procedure. Indeed, it is also possible to design metamaterials by choosing microstructures with different geometric shapes or materials. In the revised supplementary information, we have provided new candidates for metamaterial microstructures that can realize the proposed coupled resonance. The geometric parameters of the metamaterial microstructure with another geometric shape or at another target frequency can be obtained by performing the same shape optimization technique presented in the main text.

To convey this content to readers, we have added the following sentences in the revised manuscript and supplementary information:

In the revised manuscript (newly added):

(Page 11) “The proposed single-phase microstructure in Fig. 2a has the advantage of being easy to fabricate while effectively realizing a non-zero . However, it is also possible to design metamaterial microstructures with different geometries or materials (see Supplementary Fig. 6 for other microstructure candidates). In addition, we choose aluminum as the target isotropic background material and 100 kHz as the target resonance frequency in Figs. 2b-2e. However, the proposed coupled resonance is applicable to other materials and frequencies (see Supplementary Note 6 for the applicability and limitation of the proposed method). Supplementary Table 1 provides the required physical properties of the anisotropic medium to realize the coupled resonance at various target materials and frequencies. The geometric parameters of the metamaterial microstructure with another geometric shape or at another target frequency can be obtained by utilizing the same shape optimization algorithm in Ref. [28].”

Supplementary Fig. 6 (newly added)

Response to the Reviewer #2

General Comment: This study summarizes theoretical and experimental results of an ingenious mechanism for utilizing a metamaterial with appropriate bulk elastic parameters, coupled with an anisotropic Fabry-Perot resonance cell, to generate circularly polarized elastic waves at ultrasonic frequencies (e.g, 100 kHz). The presentation is well-articulated and the results are appropriately demonstrated and supported experimentally. Although interest in generating circularly polarized transverse elastic wave fields is long-standing (e.g., Einspruch, 1964), this methodology appears to me to be a significant advance in this area and has good potential to advance ultrasonic technology for material testing and other purposes. I did not find any errors in the theoretical or experimental presentation and believe that the description of the apparatus and metamaterial design (i.e., drilled aluminum) is sufficient so that the results could be reproduced and extended.

General Response: Thank you for supporting the publication of our paper in Nature Communications. Detailed responses to each comment are provided below.

Comment #1: The study gives short-shrift to the considerable work in the seismological community regarding shear-wave splitting at the mantle and crustal scale. In these circumstances the birefringence of Earth materials due to bulk mineral alignment and/or stress-related cracks creates transient elliptical or circular polarizations that have been studied for decades.

Response: To reflect the reviewer's comment, we have added an explanation of the study of circularly polarized seismic waves in geophysics and three related review papers, Refs. [4-6], to the references. Thank you for enriching our literature review.

In the revised manuscript:

(Page 2) “Since the first identification of shear-wave splitting due to the seismic birefringence of the Earth's crust [2] and upper mantle [3] in 1980, the observation of elliptically or circularly polarized elastic waves in the anisotropic Earth has been studied for decades [4-6]. In addition, some researchers have reported the realization of circularly polarized elastic waves in metals at low temperatures [7] and

artificially structured chiral materials [8-12]. Despite these efforts, the utilization of circularly polarized elastic waves is still difficult due to the lack of a method to effectively generate elastic circular polarization in common isotropic solids, which constitute most load-carrying elements in engineering applications.”

- [2] Crampin, S. et al. Observation of dilatancy-induced polarization anomalies and earthquake prediction. *Nature* **286**, 874-877 (1980)
- [3] Ando, M., Ishikawa, Y. & Wada, H. S-wave anisotropy in the upper mantle under a volcanic area in Japan. *Nature* **286**, 43-46 (1980).
- [4] Savage, M. K. Seismic anisotropy and mantle deformation: What have we learned from shear wave splitting? *Rev. Geophys.* **37**, 65-106 (1999)
- [5] Crampin, S. & Chastin, S. A review of shear wave splitting in the crack-critical crust. *Geophys. J. Int.* **155**, 221-240 (2003)
- [6] Crampin, S. & Peacock, S. A review of shear-wave splitting in the compliant crack-critical anisotropic Earth. *Wave Motion* **41**, 59-77 (2005)

Comment #2: The accompanying supplemental materials were apparently incomplete in the sense that the video demonstration/supplemental movies were only (for me) downloadable as a static Powerpoint -- I was looking forward to seeing these.

Response: We are sorry that we submitted the supplementary movies in a static PowerPoint format. We have resubmitted the revised supplementary movies in a dynamic mp4 format. Thank you for looking forward to our material, and we hope it works well this time.

REVIEWERS' COMMENTS

Reviewer #1 (Remarks to the Author):

The comments have been addressed well, I think that the manuscript can be accepted.